# GABAergic and Glutamatergic Phenotypes of Neurons Expressing Calcium-Binding Proteins in the Preoptic Area of the Guinea Pig

**DOI:** 10.3390/ijms23147963

**Published:** 2022-07-19

**Authors:** Krystyna Bogus-Nowakowska, Anna Robak, Daniel Kalinowski, Anna Kozłowska, Maciej Równiak

**Affiliations:** 1Department of Animal Anatomy and Physiology, Faculty of Biology and Biotechnology, University of Warmia and Mazury in Olsztyn, pl. Łódzki 3, 10-727 Olsztyn, Poland; ankar@uwm.edu.pl (A.R.); daniel.kalinowski@uwm.edu.pl (D.K.); mrowniak@uwm.edu.pl (M.R.); 2Department of Human Physiology and Pathophysiology, School of Medicine, University of Warmia and Mazury in Olsztyn, Warszawska 30, 10-082 Olsztyn, Poland; kozlowska.anna@uwm.edu.pl

**Keywords:** immunofluorescence, VGAT, VGLUT, calcium binding proteins, preoptic area, guinea pig

## Abstract

The mammalian preoptic area (POA) has large populations of calbindin (CB), calretinin (CR) and parvalbumin (PV) neurons, but phenotypes of these cells are unknown. Therefore, the question is whether neurons expressing CB, CR, and/or PV are GABAergic or glutamatergic. Double-immunofluorescence staining followed by epifluorescence and confocal microscopy was used to determine the coexpression patterns of CB, CR and PV expressing neurons with vesicular GABA transporters (VGAT) as specific markers of GABAergic neurons and vesicular glutamate transporters (VGLUT 2) as specific markers of glutamatergic neurons. The guinea pig was adopted as, like humans, it has a reproductive cycle with a true luteal phase and a long gestation period. The results demonstrated that in the guinea pig POA of both sexes, ~80% of CB+ and ~90% of CR+ neurons coexpress VGAT; however, one-fifth of CB+ neurons and one-third of CR+ cells coexpress VGLUT. About two-thirds of PV+ neurons express VGAT, and similar proportion of them coexpress VGLUT. Thus, many CB+, CR+ and PV+ neurons may be exclusively GABAergic (VGAT-expressing cells) or glutamatergic (VGLUT-expressing cells); however, at least a small fraction of CR+ cells and at least one-third of PV+ cells are likely neurons with a dual GABA/glutamate phenotype that may coexpress both transporters.

## 1. Introduction

Calcium binding proteins (CaBPs) are used as selective markers to identify discrete neuronal populations and pathways in the central nervous system [1]. CaBPs such as calbindin (CB), calretinin (CR) and parvalbumin (PV) are useful markers of distinct subpopulations of GABAergic neurons, e.g., in the cerebral cortex [2], hippocampus [3] and amygdala [4]. Although CB-, CR- and PV-containing cells are commonly considered as inhibitory interneurons containing GABA, not all of them use GABA [5,6], and not all of them are interneurons [7,8]. In the rat basal forebrain (BF), most PV+ neurons are GABAergic [9], and PV functions there as a marker for a subpopulation of large GABAergic neurons. These GABA/PV neurons in the rat BF have a prominent depolarizing decline during hyperpolarizing current pulses, which may be important for generating rhythmic firing, and these cells are cortically projecting [10]. On the other hand, the vast majority of cells containing CB or CR in the BF are glutamatergic, and only a minority of them are GABAergic [9]. CB and CR also appear in excitatory principal neurons in the rat and guinea pig mammillary complex nuclei, and they project to the septum or to the anterior thalamic nuclei [5,11]. Moreover, approximately half of the basal forebrain PV+ and a significant proportion of cells containing CB or CR were also immunopositive for phosphate activated glutaminase (PAG), the enzyme used in the synthesis of the glutamate, indicating that a proportion of these cells may have the potential to synthesize this neurotransmitter [9]. As some glutamic acid decarboxylase (GAD) neurons may also contain PAG, some BF neurons may have the capacity to synthesize GABA and glutamate [12]. In the guinea pig, CaBPs have been used as important markers for distinguishing subpopulations of neurons and pathways in adult and developing brains [13,14,15]. For example, these proteins mark the vast majority of GABAergic neurons in the guinea pig amygdala and subicular complex [4,14]. CR functions as a marker of immature neurons during guinea pig brain development, and it may be used for defining fetal stages in the septum and anterior thalamic nuclei [16,17], whereas CB and PV function as markers of postnatal stages during development of the guinea pig anterior thalamic and septum, respectively [16,17].

The mammalian preoptic area (POA) has large populations of CB-, CR- and PV-containing neurons; however, the phenotypes of these cells are as still obscure. The mammalian POA is not a uniform structure, and it is usually subdivided into some parts: the medial preoptic area (MPA), the lateral preoptic area (LPA), the median preoptic nucleus (MPN), and the periventricular preoptic nucleus (PPN) [18,19,20]. Neurons expressing CaBPs in the POA create separate populations, as CB and CR are coexpressed only in a small number of neurons and CB+ cells never coexpress PV [15]. Moreover neurons expressing CB in the POA are sexually dimorphic, favoring males [15,21,22,23,24,25]. The preoptic area is important for endocrine activity and essential for the expression of sexual behavior [26]. As it receives inputs from diverse brain regions that convey sensory and autonomic signals relevant to reproduction [27], the region is crucial for integrating hormonal and environmental signals and communicating them to gonadoliberin (GnRH) neurons. In the POA, especially the PPN and MPN provide the majority of GnRH-containing neurons [28], which induce the pituitary to release follicle-stimulating hormone (FSH) and luteinizing hormone (LH) in the hypothalamic pituitary–gonadal axis [29]. GABA and glutamate are predominant transmitters of the hypothalamus, and GABAergic and glutamatergic transmissions are important for both the negative and positive feedback mechanism on the GnRH release [30,31]. Induction of LH surge release requires suppression of GABA and stimulation of glutamate release into the POA [32]. Glutamate is the major fast excitatory neurotransmitter that controls not only the neuroendocrine system but also other hypothalamic regions [31], whereas GABA plays an inhibitory role in the regulation of GnRH release in the preoptic area/anterior hypothalamic region; however, GABA may have also a stimulatory effect on gonadotrophin secretion [33]. Moreover, GABAergic and glutamatergic neurons of the rat anteroventral periventricular nucleus are likely targets of estradiol and central to the generation of GnRH and LH surge release [34]. These neurons responsible for communicating signals to GnRH neurons express both vesicular glutamate transporter 2 (VGLUT2), a marker of hypothalamic glutamatergic neurons, as well as GAD and vesicular GABA transporter (VGAT), markers of GABAergic neurons [34]. 

Although the mammalian POA has a large population of CB-, CR- and/or PV- expressing neurons on the one hand, and a large population of GABAergic and/or glutamatergic cells on the other, nothing is known about the relationships among all these populations. Thus, the present study investigated for the first time the co-localization pattern of CB, PV or CR with VGAT or VGLUT2 in the POA of the guinea pig using double-immunofluorescence techniques as well as epifluorescence and confocal microscopy. Although VGLUT2 and VGAT are present in synaptic vesicles, they have also been shown to be markers of glutamatergic and GABAergic neurons [35,36,37,38,39,40]. It should be pointed out that vesicular glutamate transporters were the first definitive markers of glutamatergic neurons, and they are highly useful and reliable tools for expanding current knowledge of glutamatergic systems in CNS circuitry [38,39]. VGLUT2 has been demonstrated as a specific marker of hypothalamic glutamatergic somata as evidenced by in situ hybridization [38,40,41], northern blot analysis [38], mRNA expression [38,39] and immunohistochemistry [42,43,44]. VGAT was demonstrated in GABAergic perikarya by immunohistochemistry in the hypothalamus of mice and rats [45,46,47]. Therefore, the present study used VGLUT 2 to identify the glutamatergic nature of the POA cells, whereas VGAT was used to assess the GABAergic phenotype of the POA neurons.

As the mammalian POA is essential for the expression of sexual behavior and important for endocrine activity [26], the guinea pig was adopted as an animal model. The guinea pig has, similarly to humans, a relatively long gestation, and neonates have a highly developed central nervous system [48,49]. Moreover, in contrast to other laboratory rodents (rats or mice), it has a reproductive cycle with a true luteal phase [50]. In addition, the guinea pig displays several key physiological characteristics of gestation that more closely resemble human pregnancy, and the major advantage of the guinea pig seems to be its similarity to the human regarding maternal serum progesterone concentrations in late gestation [51]. To make the text easier to read, neurons expressing CB, PV, CR, VGAT and VGLUT2 will be uniformly described as CB+, PV+, CR+, VGAT+ and VGLUT+ neurons.

## 2. Results

### 2.1. CaBPs Immunoreactivity in the Preoptic Area of the Guinea Pig

Among all of the studied proteins, CB showed the highest density, CR lower and PV the weakest (present study and [15]). CB+ and CR+ neurons were the most abundant in the PPN and MPN, in the MPA the number of these cells was moderate, and in the LPA, only a few CB+ and CR+ neurons were present. PV+ neurons were the most abundant in the MPA, whereas in the remaining POA parts, they were single. CB+ neurons had mostly oval or triangular somata (Figure 1A,B and Figure 2A,B). The CB-immunoreactivity (CB-IR) was usually observed in the neuronal cytoplasm, nucleus and initial sections of the dendrites. In addition, it was also present in numerous intensely stained punctae similar to axon terminals. The CB-IR fibers were varicose, sometimes smooth, of variable thickness, and they created a dense fiber network, especially in the PPN and MPN. The CR staining was observed in somata, nuclei, delicate varicose or smooth fibers and punctuate structures; however, the neuropils showed much weaker immunoreactivity than that in the CB staining (Figure 3A,B and Figure 4A,B). PV+ neurons had oval or elongated somata, and they were bigger than those of CB+ or CR+ ones (Figure 5A,B and Figure 6A,B). PV+ fibers of variable length were usually smooth; however, short varicose fibers also occurred. They sometimes wrapped around the immunonegative somata, creating basket-like structures.

### 2.2. VGAT and VGLUT Immunoreactivity in the Preoptic Area of the Guinea Pig

VGAT+ and VGLUT+ immunostainings were abundantly present in all parts of the POA. VGAT+ immunoreactivity consisted of perikarya, dendrites and punctate structures (Figure 1, Figure 2, Figure 3, Figure 4, Figure 5 and Figure 6A’). VGAT+ neurons were of different size, small or relatively big, with rounded, oval or multipolar somata, and the immunoreactivity was usually confined to the cell periphery. VGAT+ dendrites were of two types: short and usually thicker, or longer and thinner (smooth or varicose) (Figure 2A’). Neuropil was stained very strongly, and the punctate structures were abundantly placed in the background. VGAT+ terminals and puncta surrounded unstained neurons, creating baskets and cartridges. VGLUT+ immunoreactivity comprised perikarya, dendrites and punctate structures (Figure 1, Figure 2, Figure 3, Figure 4, Figure 5 and Figure 6B’). The perikarya were of medium size or big, and the staining was mostly present at the cell periphery. VGLUT+ dendrites were mostly thin and varicose (Figure 2B’). The neuropil immunostaining was very strong in all POA parts and terminals, and varicose fibers touched unstained neurons.

### 2.3. The Coexpression Pattern of CB+, CR+ and PV+ Neurons with VGAT and/or VGLUT in the Preoptic Area of the Guinea Pig

All details concerning the colocalization of CB+, CR+ and PV+ neurons with VGAT and/or VGLUT in the preoptic area of the guinea pig are shown in Table 1. 

In both male and female subjects, approximately 80% of CB+ neurons contained VGAT, and about 19% of them coexpressed VGLUT (Table 1). Thus, the population of CB+ neurons was rather composed of two distinct neuronal subpopulations: CB+/VGAT+ and CB+/VGLUT+ (Figure 1A–B”). However, the existence of CB+ neurons that coexpress both VGAT and VGLUT also could be excluded. 

Most CR+ neurons coexpressed VGAT (Figure 3A” and Figure 4A”), but many of them also contained VGLUT (Figure 3B” and Figure 4B”), as in both sexes approximately 88% of CR+ cells were positive for VGAT, and approximately 30% of CR+ cells contained VGLUT (Table 1). Thus, although CR+ neurons may be solely GABAergic or solely glutamatergic, at least a proportion of them must contain both transporters, i.e., dual-phenotype VGAT+/ VGLUT+ neurons.

The population of PV+ neurons was the most diversified. Approximately 70% of them coexpressed VGAT (Figure 5A” and Figure 6A”), and about 65% coexpressed VGLUT (Table 1, Figure 5B” and Figure 6B”). Thus, the population of PV+ neurons should contain the highest percentage of dual-phenotype VGAT+/VGLUT+ neurons.

## 3. Discussion

The present study characterized, for the first time, the chemical phenotype of CB+, CR+ and PV+ neurons in the preoptic area of the guinea pig. The results show that in the guinea pig POA in both sexes, about 80% of CB+ neurons and almost 90% of CR+ neurons coexpress VGAT. However, about one-fifth of CB+ cells and one-third of CR+ cells also coexpress VGLUT. Furthermore, about two-thirds of PV+ neurons coexpress VGAT, and approximately the same proportion of PV+ neurons coexpress VGLUT. Thus, the present results revealed that many CB+, CR+ and PV+ neurons may be exclusively GABAergic (VGAT-expressing) or glutamatergic (VGLUT-expressing); however, at least a small proportion of CR+ cells and at least one-third of PV+ cells are likely dual-phenotype GABA/glutamate neurons that contain both transporters.

The results of the present study revealed that in the guinea pig POA of both sexes, large numbers of CB+ and CR+ neurons coexpress VGAT, but about one-fifth of CB+ cells and one-third of CR+ cells also coexpress VGLUT. Furthermore, about two-thirds of PV+ neurons coexpress VGAT, and approximately the same proportion of PV+ neurons coexpress VGLUT. Thus CB+ and CR+ cells of the guinea pig POA are mostly GABAergic, similar to the mouse POA, as Moffitt and co-workers [52] confirmed that the mouse POA is mostly GABAergic, containing four times more GABAergic than glutaminergic neurons. The populations of CB+, CR+ and PV+ neurons in the POA differ significantly from those in the cortex-like regions where CB, CR and PV are commonly considered as markers of GABAergic interneurons. For example, CB+, CR+ and PV+ interneurons are the main component of non-overlapping populations of GABA cells in the mammalian cerebral cortex [53,54,55,56]. In the cortex and cortex-like regions, many CB+ cells are double-bouquet interneurons, while CR appears in double-bouquet cells as well as in bipolar and Cajal-Retzius cells [57,58]. CB+ and CR+ interneurons create synapses on dendrites of pyramidal neurons and are categorized as non-fast-spiking [59]. PV is indicated in basket and chandelier populations of GABAergic neurons, and PV+ neurons create synapses on somata and axons of pyramidal neurons and are categorized as fast-spiking [60]. It should be kept in mind however, that even in the cortex-like regions where these calcium-binding proteins function as suitable markers for identifying inhibitory interneurons containing GABA, not all of these cells use GABA [5,6], and not all of them are interneurons [7,8]. For example, in rodents, almost 100% of CR+ neurons in the neocortex seem to use GABA as a neurotransmitter [2,61,62], but in the monkey [63] and human [64] neocortex, about 25% of CR+ neurons do not express GABA. Moreover, some neurons containing CB, CR or PV in the rat amygdala do not contain GAD [6]; they project to extrinsic brain regions [7,8], and a fraction of CB+ and CR+ neurons are glutamatergic projecting neurons [6,8,65]. Moreover, in the ‘‘cortex-like’’ nuclei of the amygdala, most projection neurons use glutamate as a fast, excitatory neurotransmitter [66], whereas in the ‘‘striatum-pallidum-like’’ nuclei of the amygdala, many projection cells use GABA as a fast, inhibitory neurotransmitter [67,68,69]. In the ‘‘cortex-like’’ nuclei, GAD is expressed in interneurons [70,71], whereas in the ‘‘striatum-pallidum-like’’ nuclei, it is often expressed in projection neurons [67,68,69]. In brain areas other than cortex-like regions, phenotypes of neurons containing these proteins may be completely different. For example, in the rat BF, most PV+ neurons are GABAergic [9], and PV functions as a marker for a subpopulation of large GABAergic neurons. These GABA+/PV+ neurons in the BF (many of them were distributed in the preoptic region) have a prominent depolarizing decline during hyperpolarizing current pulses, which may be important for generating rhythmic firing, and these cells are cortically projecting [10]. On the other hand, most cells containing CB or CR in the BF are glutamatergic, and only a minority of them are GABAergic [9]. CB and CR also appear in excitatory principal neurons in the rat and guinea pig mammillary complex nuclei, and they project to the septum or to the anterior thalamic nuclei [5,72]. A population of the extrinsic CR axons originating in the supramammillary area are aspartate/glutamatergic, and these axons form asymmetric synapses with septal CB+ neurons and a population of septal cholinergic cells [5]. 

The results of the present study also revealed that although in the guinea pig POA of both sexes, a large majority of CB+, CR+ and PV+ neurons express either VGAT or VGLUT2 (exclusively GABAergic or glutamatergic cells), at least some of them are supposed to contain both VGAT and VGLUT2 (dual-phenotype GABA/glutamate cells). Although direct comparisons of these results with previous studies are not possible due to the lack of similar data, they coincide well with various anatomical and physiological studies in the POA that are worth discussing.

For example, POA functions are strongly affected by actions of gonadal steroids [73] as it regulates sexual behaviors, and this region is especially rich in neurons expressing estrogen receptors alpha (ERα), estrogen receptors beta (ERβ), as well as androgen receptors [74,75,76,77]. Thus, GABAergic, glutamatergic and dual-phenotype neurons of the guinea pig POA reported in the present study are highly likely to express ERs. Interestingly ERα+ neurons constitute about 50% of MPA neurons in mice [78], and in the monkey hypothalamus CB+ neurons coexpress ERα [76]. Moreover GABA and glutamate neurons in the mice preoptic region express ERα with larger proportions of ERα neurons that are GABAergic compared with glutamatergic [78,79]. The glutamate neurons expressing ERα are essential for both the negative and positive estradiol feedback loops, whereas GABA neurons expressing ERα are only required for estradiol-positive feedback [79]. GABAergic and glutamatergic transmissions are important for both the negative and positive feedback mechanism on the GnRH release [30,31]. GnRH neurons activate the pituitary to release FSH and LH in the hypothalamic pituitary–gonadal axis, and estradiol-dependent activation of GnRH neurons induces ovulation [80]. Although GnRH neurons do not themselves express ERs [81], estrogen is one of the most important determinants of GnRH neuron activity [82] and exerts a negative feedback action upon the activity of the GnRH neurons; this results in restrained LH secretion in all mammals, including humans [83,84,85]. Induction of LH surge release requires suppression of GABA and stimulation of glutamate release into the POA [32].

It is generally expected that the inhibitory neurotransmitter GABA and the stimulatory neurotransmitter glutamate are released from different neurons, and this assumption agrees to some extent with our results, as in the guinea pig POA, most CB+, CR+ and PV+ neurons are exclusively GABAergic or glutamatergic. However, it should be emphasized that these results also revealed that at least a small portion of CR+ cells and at least about one-third of PV+ cells should be dual-phenotype GABA/glutamate neurons; thus, some POA neurons are simultaneously GABAergic and glutamatergic. It should be noted that some authors accept that neurons in the hypothalamus are exclusively either GABAergic or glutamatergic [86,87], whereas other authors conversely consider the possible existence of GABA-glutamatergic neurons. For example, Romanov and co-workers [88], using single-cell RNA sequencing, revealed in the hypothalamus subpopulations of GABAergic, glutamatergic and dopaminergic expressing genes enabling “alternative neurotransmission” or “dual neurotransmission”, which may lead to the conclusion that “single-neurotransmitter” criteria of neuronal classification are questionable. Moreover, Moffitt and co-workers [52] performed scRNA-seq plus MERFISH in the mouse preoptic region and reported some hybrid neurons releasing both GABA and glutamate. Furthermore in the rat [89], about 75% of LPA neurons that project to the lateral habenula (LHb) expressed VGluT2 mRNA, a smaller subpopulation of the LPA neurons projecting to the LHb expressed GADs mRNA, and very few—about 3.5%—coexpressed both transcripts. Such dual-phenotype neurons were also reported in the anteroventral periventricular region (AVPV) of the rat where nearly all neurons express VGLUT2 as well as glutamic acid decarboxylase and VGAT [34]. The AVPV region of the rat is the medial region of the POA previously referred to as the medial preoptic nucleus, which corresponds to the PPN and MPN nuclei of the guinea pig. Few such dual-phenotype neurons were also found in the adjacent POA region of the rat, and none were detected in either the rat hippocampus or cortex [34]. Thus, it seems likely that these dual-phenotype neurons are important for a function unique to the POA region. On the other hand, striatal efferents may also contain both glutamate and GABA [90], and the hippocampus axon terminals contain both VGLUT2- synaptic vesicles and vesicles that carry vesicular GABA transporter [91]. These dual-phenotype GABA/glutamate CR+ or PV+ neurons of the POA may integrate hormonal and photoperiodic signals necessary for LH surge release and ovulation [34]. VGAT declines and VGLUT2 rises in dual-phenotype terminals contact GnRH neurons during the afternoon [34], and such timing of the changes is necessary for the LH surge [92]. These changes may be triggered, at least in part, by neurons of the suprachiasmatic nucleus that convey photoperiodic signals to steroid-responsive neurons in the POA, which are crucial for integrating hormonal and environmental signals and communicating them to GnRH neurons that activate the pituitary to release FSH and LH in the hypothalamic pituitary–gonadal axis [93,94].

As induction of LH surge release requires suppression of GABA and stimulation of glutamate release into the POA, the question arises as to how a photoperiodic signal simultaneously activates glutamate release and inhibits GABA release from the POA neurons. The first possibility arises from the fact that most CB+, CR+ and a large population of PV+ neurons in the guinea pig POA are solely GABAergic or solely glutamatergic; the same afferent signal may have opposite effects on these GABAergic and glutamatergic populations. Moreover, on the other hand, independent LPA glutamate and GABA neurons synapse together and release either glutamate or GABA or both onto the same lateral habenula neuron providing convergent neurotransmission, thus exerting bivalent control over single lateral habenula neurons and driving opposing motivational states [89]. The second possibility is that GABA and glutamate are released from the same neurons and reciprocal release might be accomplished by photoperiodic signals and autofeedback mechanisms that are regulated by estrogen, as GABAergic and glutamatergic neurons coexpress ERα [79]. Such a possibility is supported by the evidence that terminals containing both VGAT and VGLUT2 contacted GnRH neurons [34]. These contacts were observed in the preoptic region on all medial GnRH neurons and on few lateral GnRH neurons, and the majority of medial, but not lateral, GnRH neurons contain functional NMDA receptors [95], a glutamate receptor subtype that regulates GnRH synthesis [96], critically important in the physiological regulation of gonadotropin secretion [97]. The mechanisms controlling the switch from GABA to glutamate release remain to be determined; however, the fact that glutamate is converted by GAD to GABA suggests one possibility. It is suggested that in the presence of estradiol, the daily signal to AVPV neurons in the rat inhibits GAD67 synthesis, thereby decreasing utilization of glutamate for GABA synthesis and providing more glutamate for release [34]. On the other hand, other questions concerning such a colocalization require an answer. For example, are GABA and glutamate co-released from these neurons simultaneously and from the same locus, or are they released spatially and temporally separated? What is the proportion in one terminal between synaptic vesicles containing each individual neurotransmitter? What other co-factors are possible? Furthermore, GABA, commonly known to play an inhibitory role in the regulation of GnRH release in the adult preoptic area/anterior hypothalamic region, during development, has a stimulatory effect on gonadotrophin secretion [33], as early in development, GABA is excitatory and induces depolarization of neurons [98] and switches to being inhibitory before birth [99]. Moreover, considering GABA as purely inhibitory or excitatory is to some extent an oversimplification, as GABA can act through the GABAA receptor to exert both depolarizing and hyperpolarizing effects on GnRH neurons [100]. Moreover, colocalization of an inhibitory and an excitatory neurotransmitter in one neuron or in one axon terminal may not be uncommon, as neurons releasing GABA and glutamate were reported in the hypothalamus [52,88], and such a coexistence is considered as a novel characteristic of the hypothalamus with great functional importance [88]. The function of the glutamine-glutamate/GABA cycle in the brain to transport glutamine from astrocytes to neurons and the neurotransmitter glutamate or GABA from neurons to astrocytes is a well-known concept. Glutamate (Glu) is produced from the tricarboxylic acid cycle intermediate 2-oxoglutarate by reversible reductive amination with either ammonium or glutamine as the nitrogen source. Moroz and co-workers [101] suggest that GABA may be a perfect “choice” to balance the potential overexcitation/neurotoxicity induced by glutamine. GABA is produced from Glu and can be a conserved evolutionary solution for Glu inactivation or reduction of its concentrations. At the same time, GABA can also fuel the tricarboxylic acid cycle, recovering Glu as a by-product. Therefore, these dual-phenotype GABA/glutamate neurons of the guinea pig POA could be an example of the metabolic coupling between Glu-GABA, which appears to be the perfect pair for biologically and chemically differentiated signaling in neural circuits [101].

One issue more is worth discussing. In the guinea pig POA, CB+ neurons are sexually dimorphic, favoring males [15], and the present results have shown that these neurons are mostly GABAergic; however, there were no sex differences in the number of CB+/GABA+ neurons. The data are in agreement with the results for the mouse MPA, where about 80% of neurons are VGAT+, and with results obtained in the sexually dimorphic nucleus of the preoptic area (SDN-POA) of the rat and human [102]. The SDN-POA of the rat and human exhibited high levels of GAD mRNA expression, and the expression did not show sex difference in either the rat or human. Moreover, nearly all if not all neurons of the rat and human SDN-POA contain GAD mRNA, so GABA is the predominant neurotransmitter of the SDN-POA, and the output of the region is generally inhibitory. The preoptic area projects to other hypothalamic areas and to brainstem reticular formation [103,104,105] where it could participate in the regulation of a variety of homeostatic and reproductive functions. GABAergic neurons of the POA project to wakefulness-related areas in the hypothalamus and brainstem, which fire at a rapid rate during wakefulness, slow down during NREM sleep, and cease firing during REM sleep, and are implicated in the maintenance of wakefulness [106]. These neurons send projections to the lateral hypothalamic area, and they create appositions with orexin neurons, which play an important role in the maintenance of wakefulness and exhibit an excitatory influence on arousal-related neurons [106]. 

## 4. Materials and Methods

### 4.1. Animals

The study was performed on brains of sexually mature Dunkin–Hartley guinea pigs (Cavia porcellus) of both sexes (6 males and 6 females). Animals were obtained from Nofer Institute of Occupational Medicine in Łódz, Poland. Animals were maintained under standard laboratory conditions of lighting (12:12 h light:dark cycles) and were fed chow and tap water supplemented with vitamin C. All experiments were carried out according to the Local Ethical Commission for Animal Experimentation of the University of Warmia and Mazury in Olsztyn (No. 58/2014) and in accordance with the European Union Directive 2010/63/EU for animal experiments. All efforts were made to minimize animal suffering and to use the minimum number of animals necessary to produce reliable scientific data. 

### 4.2. Tissue Preparation 

Animals were anesthetized by an intraperitoneal injection of pentobarbital (Morbital, Biowet, Poland; 2 mL/kg body weight) according to the guidelines of the Humane Society Veterinary Medical Association. After cessation of breathing, they were immediately perfused intracardially with phosphate-buffered saline (PBS; pH 7.4) containing 1% sodium nitrite followed by 4% buffered paraformaldehyde (pH 7.4). Following perfusion, brains were dissected out from the skulls, post-fixed overnight in 4% paraformaldehyde, washed twice in 0.1 M phosphate buffer (pH 7.4) and then cryoprotected in sucrose solution (19%, 30%) in PBS at 4 °C until full tissue infiltration. Finally, these brains were frozen and cut into 10 µm coronal sections using a cryostat and stored at −80 °C until further processing. 

### 4.3. Immunofluorescence Experiments 

The sections containing preoptic region were processed for routine double immunofluorescence labelling using primary antisera raised in different species and species-specific secondary antibodies (Table 2). All samples were washed 3 times in PBS, incubated for 1 h with blocking buffer (10% normal goat serum, 0.1% bovine serum albumin, 0.01% NaN_3_, Triton X-100 and thimerosal in PBS) and then incubated overnight at room temperature with a mixture of primary antibodies composed of antisera to one of the CaBPs (CB, CR or PV) and a marker of GABAergic or glutamatergic neurons (VGLUT or VGAT) (Table 2). In order to show the binding sites of the primary antisera with antigens, sections were then incubated for an hour with a mixture of species-specific secondary antibodies (Table 2). The method details have been described in our previous manuscripts [15,107].

### 4.4. Controls

The specificity of the primary antibodies directed against CB, CR and PV has been shown by various researchers using these products in numerous previous studies [108,109,110,111] including our team [4,15,72,112,113]. Moreover, the mouse anti-CB (300) and mouse anti-CR (6B3) antisera included immunoblots of the guinea pig brain homogenates, which were specifically stained by these antibodies, exhibiting bands at 28 and 29 kDa, respectively (manufacturer’s technical information). The same documents also show an absence of specific immunohistochemical staining in brain sections of CB or CR knock-out mice using these antibodies. Affinity-purified rabbit antibodies directed against the vesicular glutamate transporters (Synaptic Systems, Göttingen, Germany, 135402) and anti-vesicular GABA transporter (Millipore, Burlington, MA, USA, AB5062P) were repeatedly applied in different studies [35,57,114,115,116,117,118], and their specificity has been confirmed by the manufacturers. To verify the secondary antisera specificity, omissions and replacement of primary antisera by non-immune sera or PBS were applied. A lack of immunoreactivity indicated specificity.

### 4.5. Counts and Statistics 

To evaluate the colocalizations of CB+, CR+ and PV+ neurons with VGAT or VGLUT in the guinea pig POA, selected double-labelled sections were analyzed. The numbers of single- and double-labelled CB+, CR+, and PV+ cells were manually counted. Single-labelled VGAT+ and VGLUT+ neurons were excluded from the analysis. For each combination of antigens (CB/VGAT, CR/VGAT and PV/VGAT or CB/VGLUT, CR/VGLUT and PV/VGLUT) in each animal, neurons were counted on eight evenly spaced sections arranged from the rostral to the caudal extent of the POA. The space between these sections was always 180 μm. Neurons on each section were counted at 40× magnification using a 347.6 µm × 260.7 µm region as the test frame. The test frames were set on a section in a way to cover the whole cross-section area of the POA. The neuroanatomical landmarks used to define the boundaries of the preoptic area were adopted from the atlas of the guinea pig hypothalamus [119]. In the test frames, single-labelled and double-labelled neurons were counted separately. Such separate counts made within the test frames were added together in the section and then summed up from eight sections. Finally, counts from specimens were averaged in appropriate sex and expressed as means ± standard deviation (SD). Sections were coded so that the analyses were performed without knowledge of the specimen parameters (sex, antigen, etc.). To compare the percentages of colocalized neurons in both sexes, data were analyzed by the two-tailed t-test (* *p* ≤ 0.05, ** *p* ≤ 0.01, *** *p* ≤ 0.001). The sections were analyzed using an epifluorescence microscope Olympus BX51 equipped with a CCD camera connected to a PC. Images were acquired with Cell-F software (Olympus GmbH, Hamburg, Germany). However, the CB, CR and PV signals were present mostly in cytoplasm, while the VGAT and VGLUT immunoreactivity was usually confined to the cell periphery; in many double-stained cells, the colocalization was not evident. Therefore, to conclusively ascertain that VGAT/VGLUT and CB or CR or PV co-exist in the same neurons, additional analysis was performed with the use of a confocal microscopy. The confocal fluorescent images were obtained with an LSM 700 (Zeiss, Jena, Germany) equipped with the appropriate filter for Alexa Fluor using ZEN 2009 software (Zeiss, Germany). The emission wavelengths were set up for Alexa 488 at 495 to 550 nm and for Alexa 568 at 600 to 650 nm.

## 5. Conclusions

The present results show that in the guinea pig POA of both sexes, a large number of CB+ and CR+ neurons are GABAergic, and about 20% of CB+ neurons and one-third of CR+ cells are glutamatergic. Moreover, about two-thirds of PV+ neurons coexpress VGAT, and roughly the same proportion of PV+ neurons coexpress VGLUT. Thus, CB+, CR+ and PV+ neurons may be exclusively GABAergic or glutamatergic; however, at least a small fraction of the CR+ population and at least about one-third of the PV+ population likely contain both GABA and glutamate. The large populations of CaBP neurons that are solely GABA- or glutamatergic seem to be important for both the negative and positive feedback mechanism on the GnRH release in the POA. However, the dual-phenotype GABA/glutamate neurons may integrate hormonal and environmental signals and communicate them to GnRH neurons in the presence of estrogen, inhibiting GABA release and stimulating the glutamate release that is required for the LH surge release.

## Figures and Tables

**Figure 1 ijms-23-07963-f001:**
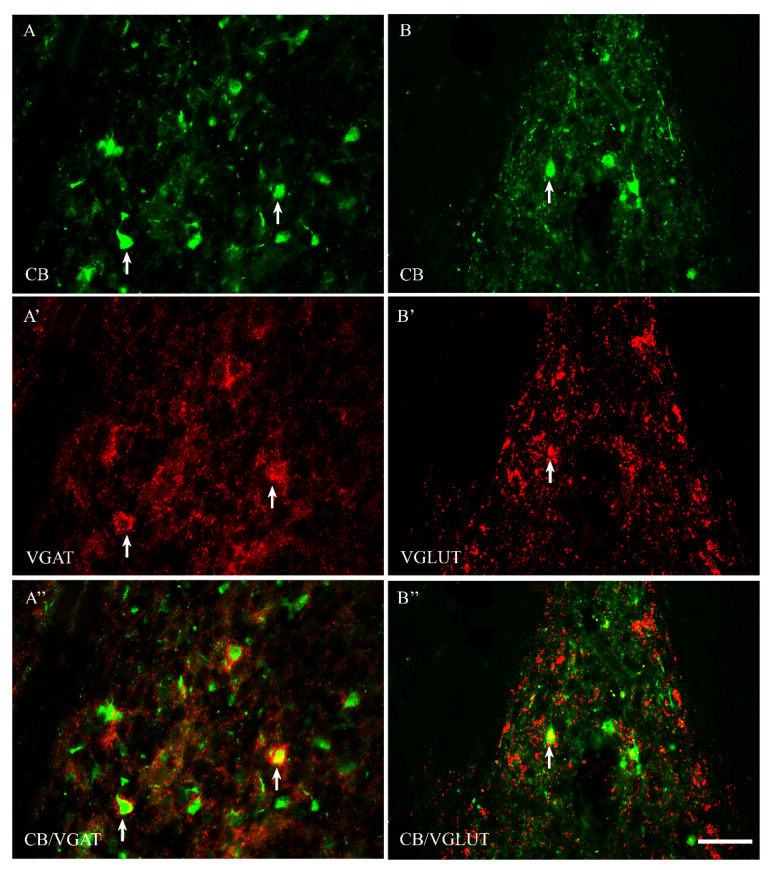
Representative microphotographs demonstrating the anatomical relationships between CB+ neurons and VGAT (**A**–**A”**) as well as CB+ neurons and VGLUT (**B**–**B”**) in the median preoptic nucleus of the guinea pig. Arrows indicate double-labelled cells. Scale bar = 50 μm.

**Figure 2 ijms-23-07963-f002:**
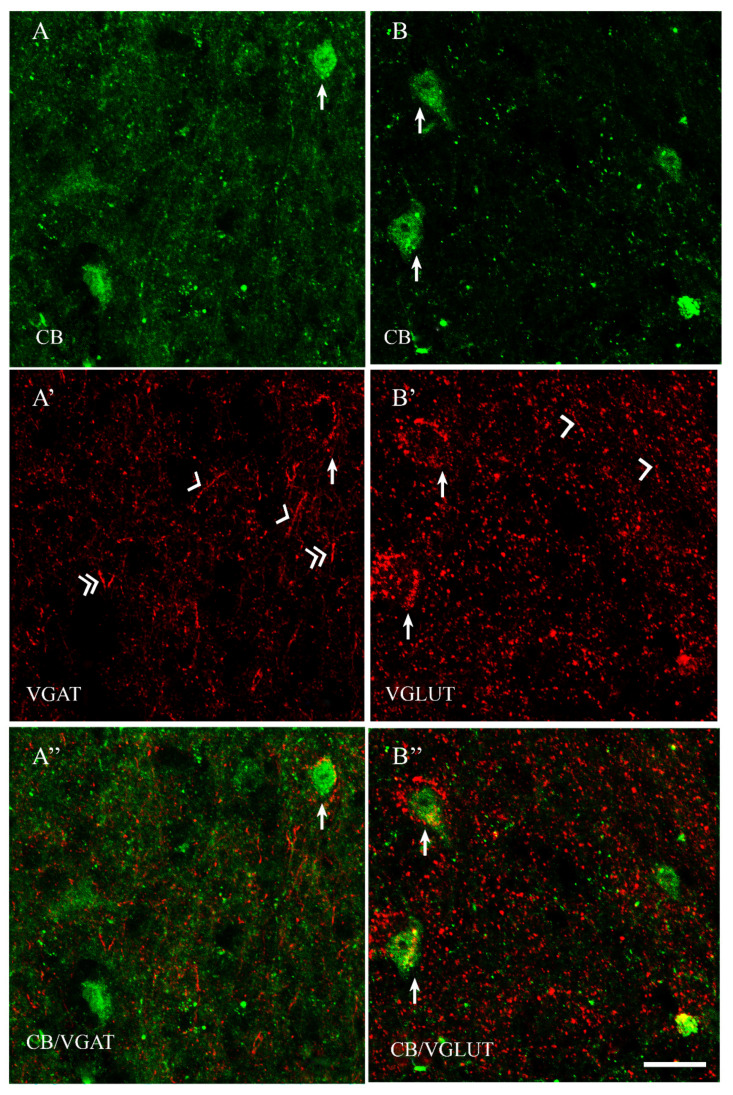
Double-labeling confocal immunofluorescence illustrating the anatomical relationships between CB+ neurons and VGAT (**A**–**A”**) as well as CB+ neurons and VGLUT (**B**–**B”**) in the periventricular preoptic nucleus of the guinea pig. Arrows show CB+ neurons that were simultaneously VGAT+ (**A**–**A”**) and/or VGLUT+ (**B**–**B”**); arrowheads indicate varicose dendrites; double-arrowheads indicate smooth dendrites. Scale bar = 20 μm.

**Figure 3 ijms-23-07963-f003:**
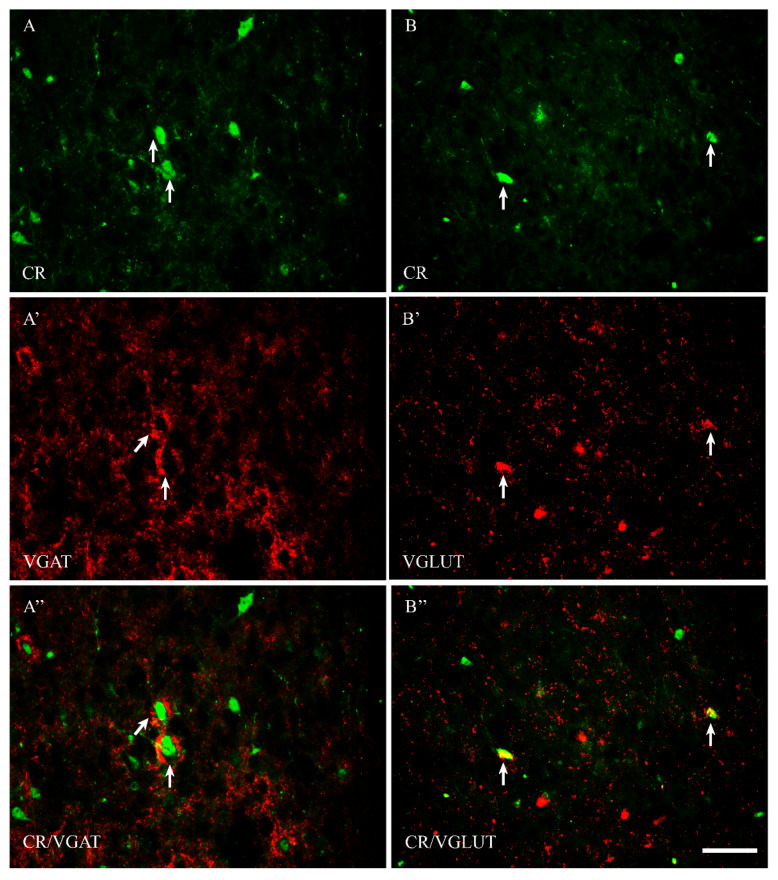
Representative microphotographs demonstrating the anatomical relationships between CR+ neurons and VGAT (**A**–**A”**) as well as CR+ neurons and VGLUT (**B**–**B”**) in the periventricular preoptic nucleus of the guinea pig. Arrows indicate double-labelled cells. Scale bar = 50 μm.

**Figure 4 ijms-23-07963-f004:**
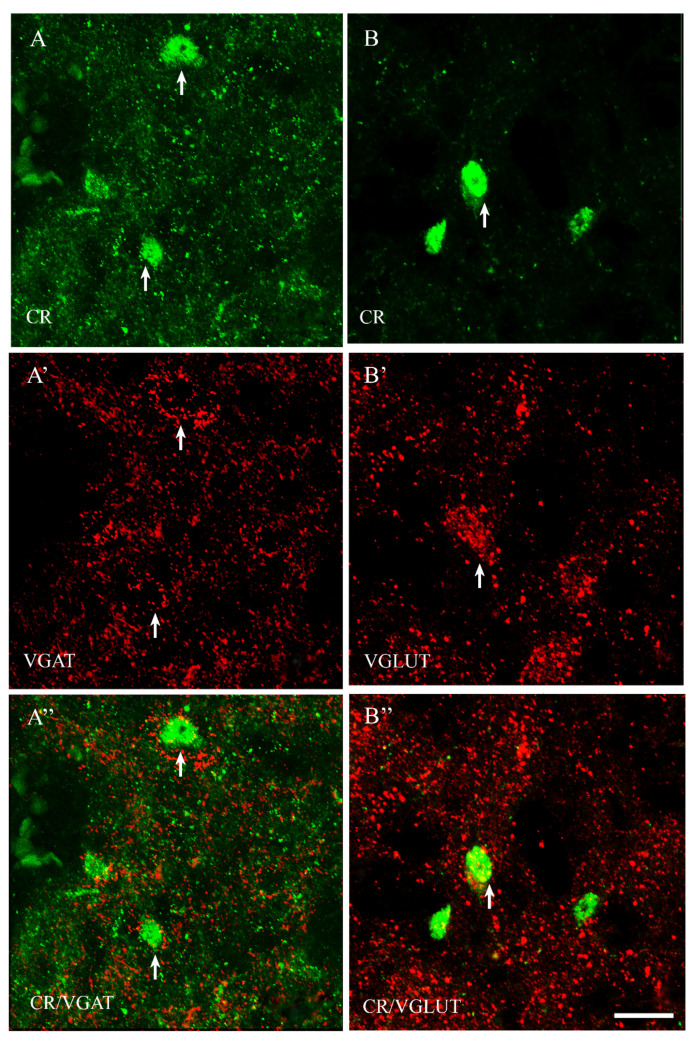
Double-labeling confocal immunofluorescence illustrating the anatomical relationships between CR+ neurons and VGAT (**A**–**A”**) as well as CR+ neurons and VGLUT (**B**–**B”**) in the median preoptic nucleus of the guinea pig. Arrows show CR+ neurons that were simultaneously VGAT+ (**A**–**A”**) and/or VGLUT+ (**B**–**B”**). Scale bar = 20 μm.

**Figure 5 ijms-23-07963-f005:**
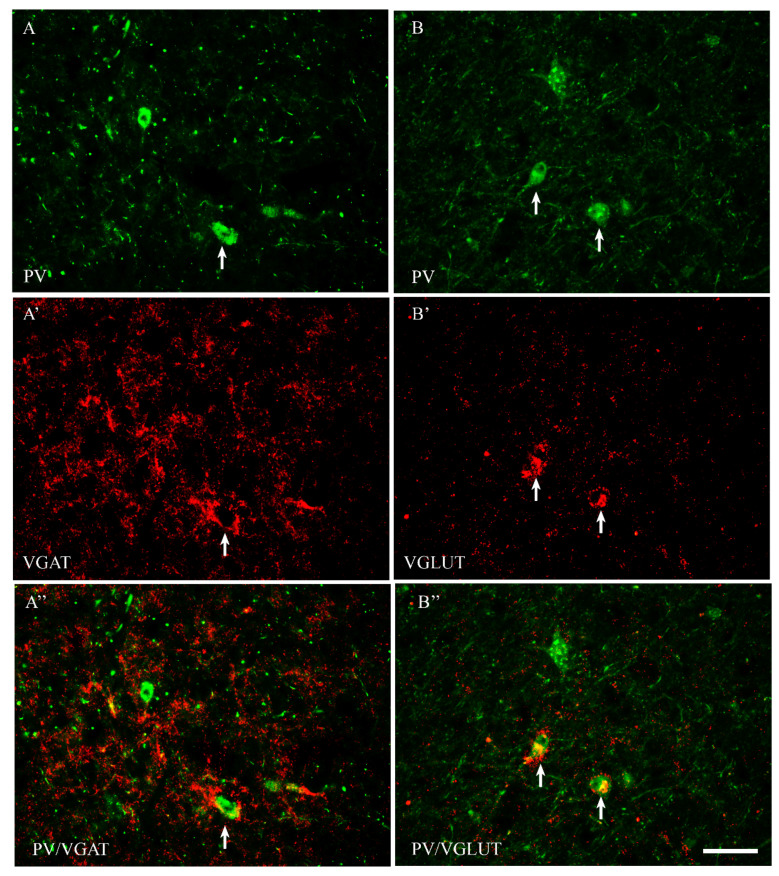
Representative microphotographs demonstrating the anatomical relationships between PV+ neurons and VGAT (**A**–**A”**) as well as PV+ neurons and VGLUT (**B**–**B”**) in the medial preoptic area of the guinea pig. Arrows indicate double-labelled cells. Scale bar = 50 μm.

**Figure 6 ijms-23-07963-f006:**
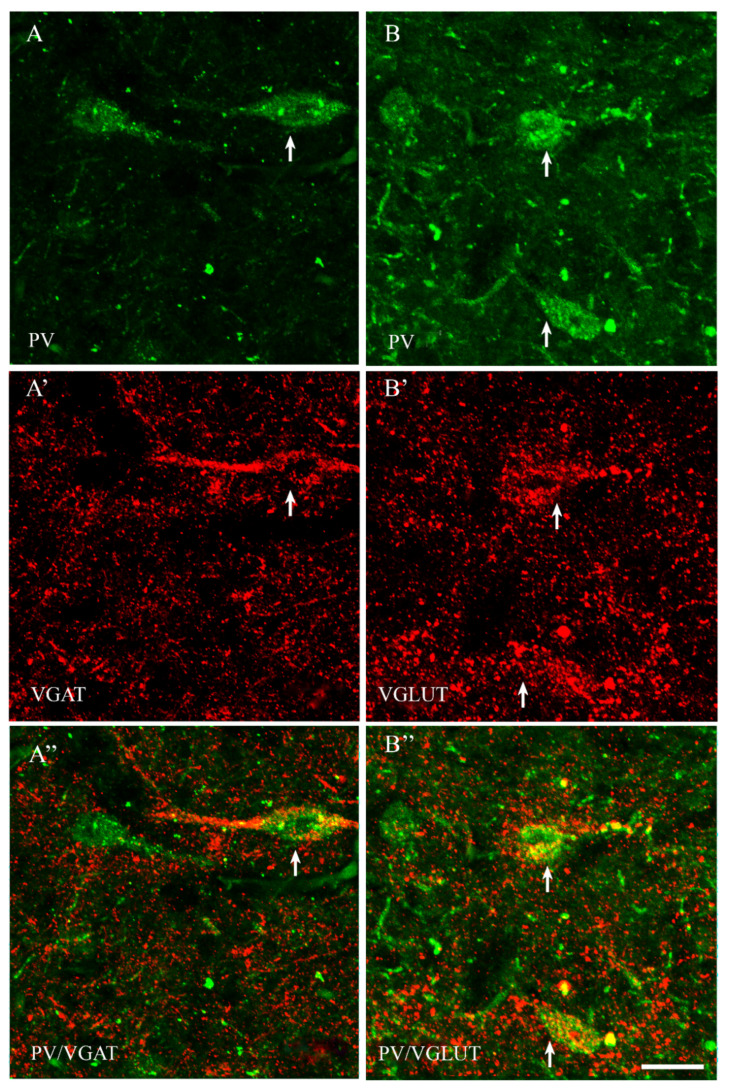
Double-labeling confocal immunofluorescence illustrating the anatomical relationships between PV+ neurons and VGAT (**A**–**A”**) as well as PV+ neurons and VGLUT (**B**–**B”**) in the medial preoptic area of the guinea pig. Arrows show PV+ neurons that were simultaneously VGAT+ (**A**–**A”**) and/or VGLUT+ (**B**–**B”**). Scale bar = 20 μm.

**Table 1 ijms-23-07963-t001:** The colocalization pattern of CB+, CR+ and PV+ neurons with VGAT and VGLUT in the preoptic area of the guinea pig. Data are presented as mean ± standard deviation (SD).

	**CB+**	**CB+/VGAT+**	**%**	**CB+**	**CB+/VGLUT**	**%**
Male	262.6 ± 3.8	208.6 ± 4.43	79.4 ± 1.8	259.3 ± 8.3	49.6 ± 3.2	19.1 ± 1.7
Female	248 ± 11.5	199 ± 7.9	80.4 ± 6.7	246.6 ± 10.6	46.6 ± 5.2	18.9 ± 1.8
	**CR+**	**CR+/VGAT+**	**%**	**CR+**	**CR+/VGLUT**	**%**
Male	157 ± 8.8	138 ± 4.9	88.2 ± 7.4	156.3 ± 9.9	50 ± 5.35	32.2 ± 5.09
Female	164 ± 5.5	145.6 ± 6.5	88.6 ± 6.4	151.6 ± 7.3	45.3 ± 6.1	30 ± 5.00
	**PV+**	**PV+/VGAT+**	**%**	**PV+**	**PV+/VGLUT**	**%**
Male	90.3 ± 4.0	62 ± 3.5	68 ± 1.1	97 ± 3.5	64 ± 4.3	65.9 ± 3.2
Female	98.6 ± 3.0	70.3 ± 2.6	71.3 ± 4.2	93 ± 6.3	60 ± 2.16	64.9 ± 6.4

**Table 2 ijms-23-07963-t002:** Specification of reagents.

Antigen	Code	Clonality	Host Species	Dilution	Supplier	Location
Primary antibodies						
CB	300	monoclonal	Mouse	1:4000	Swant	Bellinzona/Switzerland
PV	P3088	monoclonal	Mouse	1:4000	Sigma Aldrich	St. Louis, MO/USA
CR	6B_3_	monoclonal	Mouse	1:4000	Swant	Bellinzona/Switzerland
VGLUT	135 402	polyclonal	Rabbit	1:2000	SYSY	Göttingen/Germany
VGAT	AB5062P	polyclonal	Rabbit	1:2000	Millipore	Temecula, CA/USA
Secondary reagents						
ALEXA Fluor 488	A-21202	polyclonal	Donkey	1:1000	Molecular Probes	Rockford, IL/USA
ALEXA Fluor 555	A-31572	polyclonal	Donkey	1:1000	Molecular Probes	Rockford, IL/USA

## Data Availability

The data that support the findings of this study are available from the corresponding author upon reasonable request.

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
