# Peer review of "GABAergic and Glutamatergic Phenotypes of Neurons Expressing Calcium-Binding Proteins in the Preoptic Area of the Guinea Pig"

_ijms, 2022, doi:10.3390/ijms23147963_

Round 1

Reviewer 1 Report

Dear Authors,

The manuscript is original experimental research and makes an important fundamental contribution to the neurology/morphology field. 

Just one suggestion for authors: to make an additional table with abbreviations, it would be helpful for readers.

Some ideas about the dual-phenotype GABA/glutamate neurons could be found in a research paper: Moroz and co-workers were shown glutamate/GABA metabolism (Moroz et al., 2021, Evolution of glutamatergic signaling and synapses) including some ideas about converting using the same enzymes both to Glu and GABA. This paper can explain your results, and Romanov et al., 2017.

Author Response

The manuscript is original experimental research and makes an important fundamental contribution to the neurology/morphology field.

Response: I would like to express my gratefulness to the Reviewer for his/her effort and time put into the review process of the manuscript. Please find below a detailed point-by-point response to your comments. The changes made in the original text were marked in yellow.

Just one suggestion for authors: to make an additional table with abbreviations, it would be helpful for readers.

Response: According to the Reviewer suggestion, a table with abbreviations was added to the manuscript. Please see the pages 1-2.

Some ideas about the dual-phenotype GABA/glutamate neurons could be found in a research paper: Moroz and co-workers were shown glutamate/GABA metabolism (Moroz et al., 2021, Evolution of glutamatergic signaling and synapses) including some ideas about converting using the same enzymes both to Glu and GABA. This paper can explain your results, and Romanov et al., 2017.

Response: I would like to thank the Reviewer for pointing this valuable and interesting publication. Some ideas concerning the dual-phenotype GABA/glutamate neurons proposed in the paper of Moroz and co-workers (Moroz et al., 2021) were added to the Discussion. To address this concern of the Reviewer directly in the text of the revised manuscript, the following sentences were added to the Discussion (please see lines 352-364). The text is also below:

The function of the glutamine-glutamate/GABA cycle in the brain to transport glutamine from astrocytes to neurons and the neurotransmitter glutamate or GABA from neurons to astrocytes is a well-known concept. Glutamate (Glu) is produced from the tricarboxylic acid cycle intermediate 2-oxoglutarate by reversible reductive amination with either ammonium or glutamine as the nitrogen sources. Moroz and co-workers [101] suggest that GABA may be a perfect ‘choice’ to balance the potential overexcitation/neurotoxicity induced by glutamine. GABA is produced from Glu and can be a conserved evolutionary solution for Glu inactivation or reduction of its concentrations. At the same time, GABA can also fuel the tricarboxylic acid cycle, recovering Glu as a by-product. Therefore, these dual-phenotype GABA/glutamate neurons of the guinea pig POA could be an example of the metabolically coupling between Glu-GABA, which appears to be the perfect pair for biologically and chemically differentiated signalling in neural circuits [101]”.

Reviewer 2 Report

Authors have performed valuable studies. The introduction gives us deeper insight into the calcium binding proteins and the preopticarea, completed with functional relations. The experiments have been well designed and performed, the results and discussion are convincing. Hence this paper can be accepted in its present form.

Author Response

Authors have performed valuable studies. The introduction gives us deeper insight into the calcium binding proteins and the preoptic area, completed with functional relations. The experiments have been well designed and performed, the results and discussion are convincing. Hence this paper can be accepted in its present form.

Response: I would like to express my sincere thanks to the Reviewer for his effort and time put into the review process of the manuscript.
